# Identifying Potent Nonsense-Mediated mRNA Decay Inhibitors with a Novel Screening System

**DOI:** 10.3390/biomedicines11102801

**Published:** 2023-10-16

**Authors:** Julie Carrard, Fiona Ratajczak, Joséphine Elsens, Catherine Leroy, Rebekah Kong, Lucie Geoffroy, Arnaud Comte, Guy Fournet, Benoît Joseph, Xiubin Li, Sylvie Moebs-Sanchez, Fabrice Lejeune

**Affiliations:** 1Univ. Lille, CNRS, Inserm, UMR9020-U1277—CANTHER—Cancer Heterogeneity Plasticity and Resistance to Therapies, F-59000 Lille, France; 2Université de Lyon, Université Claude Bernard Lyon 1, CNRS, INSA Lyon, ICBMS, UMR 5246, Bâtiment Lederer, 1 Rue Victor Grignard, F-69622 Villeurbanne, France

**Keywords:** nonsense mutation, small molecules, therapy, nonsense-mediated mRNA decay, genetic disease

## Abstract

Nonsense-mediated mRNA decay (NMD) is a quality control mechanism that degrades mRNAs carrying a premature termination codon. Its inhibition, alone or in combination with other approaches, could be exploited to develop therapies for genetic diseases caused by a nonsense mutation. This, however, requires molecules capable of inhibiting NMD effectively without inducing toxicity. We have built a new screening system and used it to identify and validate two new molecules that can inhibit NMD at least as effectively as cycloheximide, a reference NMD inhibitor molecule. These new NMD inhibitors show no cellular toxicity at tested concentrations and have a working concentration between 6.2 and 12.5 µM. We have further validated this NMD-inhibiting property in a physiopathological model of lung cancer in which the *TP53* gene carries a nonsense mutation. These new molecules may potentially be of interest in the development of therapies for genetic diseases caused by a nonsense mutation.

## 1. Introduction

Nonsense mutations affect about 10% of patients with genetic diseases [1]. They are among the most impactful, because they very often lead to the absence of expression of the mutated gene; a quality control mechanism called nonsense-mediated mRNA decay (NMD) recognizes and rapidly degrades mRNAs carrying a premature termination codon [2,3,4,5]. This mRNA surveillance mechanism is highly regulated [6] and entwined in interactions with various metabolic pathways [7].

NMD involves different proteins, including the UPF factors UPF1, UPF2, UPF3 (also called UPF3a), and UPF3X (also called UPF3b). UPF1 has 5′ to 3′ RNA helicase activity stimulated by factors UPF2 and UPF3/3X [8,9,10]. In addition, UPF1 is a phosphoprotein requiring a cycle of phosphorylation and dephosphorylation to allow the induction of NMD on an mRNA. This phosphorylation is ensured by the SMG1 and AKT1 proteins [11,12,13].

As approximately 10% of patients suffering from a genetic disease carry a nonsense mutation responsible for the pathology, the inhibition of NMD represents an interesting therapeutic approach [3,5,14,15,16]. Since the recognition of a premature stop codon (PTC) by the NMD mechanism is similar from one tissue or organ to another, it is expected that the inhibition of NMD should be applicable to a wide variety of genetic disease cases sharing the same molecular etiology [17]. Although several dozen NMD inhibitory molecules have already been identified, they can often interfere with certain essential mechanisms of the cell, such as translation, or be toxic, like apoptosis inducers [18,19]. It is therefore essential to identify new molecules that are safe and effective, in order to study the NMD-inhibition-based therapeutic approach.

In previous studies, we screened several thousand molecules to identify NMD inhibitors [12,19,20,21,22]. The screening system used was based on targeting one of the UPF proteins in the 3′UTR part of an mRNA coding for firefly luciferase, thanks to the presence of the recognition sequences of the viral protein MS2 and to the expression of an MS2-UPF1 fusion protein. The presence of the UPF1 protein downstream of the physiological stop codon leads to the activation of the NMD of this mRNA [23]. With this screening system, apoptosis inducers [19] and cytoskeleton disruptors [22] have been shown to inhibit NMD. So have amlexanox, a drug used to treat canker sores and certain forms of asthma [21], and inhibitors of the AKT1 kinase [12]. This screening system is certainly effective, but it requires transfecting human cells with two plasmids, one expressing the gene coding for luciferase and the other expressing the gene coding for the MS2-UPF1 fusion protein. Here, in order to make identification of NMD inhibitors even more efficient, we have developed a new screening system, improved through the use of a single plasmid. We have tested this system on molecules selected by the first screening system and present here the validation of the two molecules selected in both screening systems.

## 2. Materials and Methods

### 2.1. Culture Cell

U2OS cells were grown in Dulbecco’s modified Eagle’s medium with Glutamax (Gibco, Life technologies, UK) containing 10% fetal calf serum (Sigma-Aldrich, Saint-Quentin-Fallavier, France) and 1% zellshield (Minerva Biolabs, Berlin, Germany). These cells were transfected with JetOptimus (polyplus transfection, Illkirch, France).

Calu-6 cells were grown in Roswell Park Memorial Institute medium 1640 with Glutamax, containing 10% fetal calf serum (Sigma-Aldrich) and 1% zellshield (Minerva Biolabs).

16HBE14o- cells were grown in Minimum Essential Medium Eagle with Glutamax (Gibco), containing 10% fetal calf serum (Sigma-Aldrich) and 1% zellshield (Minerva Biolabs).

### 2.2. RNA Extraction and RT-PCR Analysis

Total RNA was extracted with RNazol reagent (Molecular Research Center, Cincinnati, OH, USA), according to the manufacturer’s protocol, from about 2 million cells. RT-PCR was performed as described [21]. The primer sequences used in this study were for p53 (sense 5′-ATGTGCTCAAGACTGGCGC-3′; antisense 5′-GACAGCATCAAATCATCC-3′), GAPDH (sense 5′-CATTGACCTCAACTACATGG-3′; antisense 5′-GCCATGCCAGTGAGCTTCC-3′), and Firefly luciferase (sense 5′TGAATTGGAATCGATATTG3′; antisense 5′TTACAATTTGGACTTTCCG3′).

### 2.3. NMD Efficacy Assay

The measurement of NMD efficacy with the two-plasmid screen has been described previously in detail [21]. For the one-plasmid screening system, 180ng of a vector carrying the cDNA coding for firefly luciferase and two copies of the same intron located in the coding sequence and in the 3′UTR, are introduced into 8 million U2OS cells with JetOptimus (PolyPlus). After 24 h, the cells were distributed into 96-well white clear-bottom plates and treated with cell culture before adding the molecules for an additional 24 h. Then, the luciferase substrate Steadylite Plus (Revvity, Bussy Saint Martin, France) was added to each well before reading the luciferase activity with a Tristar luminometer (Berthold, Thoiry, France).

### 2.4. Toxicity Assay

The toxicity assay was performed 24 h after starting cell treatment in a 2 mL cell culture (6 cm dish). The cell culture supernatant was centrifuged at 6000× *g* for 5 min to pellet the cells and cell debris. The supernatant (20 µL) was then used to measure adenylate kinase activity according to the supplier’s protocol (Lonza Biosciences, Visp, Switzerland). Luciferase activity was then measured with a Tristar luminometer (Berthold).

### 2.5. Chemicals

The University of Lyon 1 chemical library screened in this work was obtained from the ICBMS, as a 10 mM solution in DMSO in 96-well plates. It consists of 3500 original synthetic or semi-synthetic compounds. Analogues of **1a** were synthesized by standard amide coupling procedures (HATU, DIPEA in DMF at r.t. for 16 h), starting from 5-nitrofuran-2-carboxylic acid (or analogues) and the corresponding amines or alcohol, or by activation of the acid (with CDI in THF) and displacement of the resulting acyl imidazole with an amine. All analogues of **2a** were already available from the ICBMS. **2f**, **2g** were prepared according to standard procedures from unprotected γ quinide lactone, obtained in two steps from D-quinic acid, and from diacetylated caffeoyl chloride, obtained in two steps from caffeic acid. Other compounds from this series were synthesized as reported previously [24,25].

## 3. Results

### 3.1. Identification of Two New Potential NMD Inhibitors

With the aim of identifying new NMD inhibitors with strong inhibitory capacity and low toxicity, an original molecule library of approximately 3500 compounds was screened with the same screening system as used previously to identify, notably, amlexanox [21]. Briefly, HeLa cells were co-transfected with a plasmid encoding the firefly luciferase mRNA and carrying MS2 protein recognition sequences and with a plasmid encoding a fusion protein composed of the viral protein MS2 and the central factor of NMD, the UPF1 protein, as previously described [12,21].

Two, among the 3500 molecules of the library, gave rise to higher luciferase activity than that promoted by G418 (used as a positive control as it had previously been shown to inhibit NMD [26,27,28]). One of the selected molecules (henceforth called **1a**) is located on plate 25 at position H11, and the other (henceforth called **2a**) is located on plate 40 at position F02 (Figure 1). We chose to study these two molecules in more detail because of their superiority over G418 in the screen, potentially indicative of a greater capacity to inhibit NMD.

### 3.2. Validation of **1a** and **2a** as NMD Inhibitors

The screening scheme used to obtain the data of Figure 1 requires the cells to be transfected with two plasmids. This leaves the possibility of cells receiving only one or the other plasmid. To remedy this problem, a new construct was generated and is depicted in Figure 2A. The plasmid carries the cDNA coding for firefly luciferase, into which an intron has been introduced in order to interrupt this open reading frame (ORF) and to make the corresponding mRNA more strongly translated [26]. To render this mRNA subject to NMD, the same intron as that introduced into the ORF was inserted more than 55 nucleotides downstream of the physiological stop codon. Consequently, this physiological stop codon became a PTC. Using the same intron in the ORF and in the 3′UTR part makes it possible to exclude molecules liable to specifically inhibit the removal of this intron through splicing, because no luciferase activity would be detected in the presence of such molecules. Thus, the higher the luciferase activity measured from this construct, the greater the inhibition of NMD. First, the splicing of the introns out of this construct was evaluated by radioactive RT-PCR for high sensitivity. Amplification was done with a primer upstream of the first intron and a primer downstream of the second intron. However, since removal through splicing of the intron in the 3′UTR portion leads to activation of NMD of the corresponding mRNA, the analysis had to be done in the presence of NMD inhibitors (Figure 2B). As expected, no amplification was detected in the absence of NMD inhibition (DMSO). This means that the second intron was very efficiently spliced out. In the presence of NMD inhibitors such as cycloheximide, amlexanox, colchicine, molecule **1a**, or molecule **2a**, two isoforms appeared [21,22]. These two isoforms are the mRNA and the isoform lacking only one intron—presumably, the second one but not the first since the former is sensitive to NMD. We cannot exclude, however, the presence of the isoform lacking the first intron but not the second. Interestingly, treatment with molecules **1a** and **2a** also led to increased levels of different firefly luciferase RNA species, which confirms their ability to inhibit NMD. These results show that the new construct is subject to NMD, since the level of the corresponding mRNA increased with the use of different NMD inhibitors. They also show that molecules 1a and 2a, like colchicine, amlexanox, and cycloheximide, inhibit NMD.

### 3.3. Derivatives of **1a** and **2a** Retain NMD-Inhibiting Activity

Once the NMD-inhibiting activity of molecules **1a** and **2a** was confirmed, it was important to initiate a structure–activity relationship (SAR) study in order to identify the chemical groups essential to this inhibitory activity. For this, several derivatives of each molecule were obtained and tested in U2OS cells for their ability to inhibit NMD of the mRNA from the firefly luciferase construct described in Figure 2A (Figure 2C and Appendix A). Each molecule was tested at 0.1, 1, 10, and 100 µM. Although no derivative was found to inhibit NMD more strongly than the original molecules, it was nevertheless possible for the 1a-family derivatives to establish a basic skeleton essential to NMD inhibition, thanks to their simpler structural features as compared to the **2a**-family derivatives (Figure 2D). A few analogues of **1a** were synthesized during this project via simple coupling reactions between a heterocyclic carboxylic acid and various amines or alcohols. Replacing the furan nucleus with a thiophene (**1h**) or a pyrazole (**1i**) led, respectively, to decreased luciferase activity or none at all. It is worth noting that certain changes in the structure of **1a** resulted in almost complete loss of activity, as observed when the nitro group was omitted (**1b**), when the amide linkage was replaced with an ester linkage (**1f**), and when an aromatic 4-hydroxy-amine was used (**1e**)**.** The trans-4-hydroxycyclohexylamine moiety was also altered in different ways. When its hydroxyl group was replaced with an amino group, the trans derivative retained some residual activity at 10 µM (**1j**), and the cis compound (**1k**) showed even less. Placing a secondary amine with a 4-aminopiperidine moiety gave rise to only low activity at 10 µM (**1c**), whereas replacing this compound with a side chain containing an amide and a dimethylamine (**1g**) partially restored the activity. Together, these results suggest that further structural modifications of **1a** should be directed toward its 4-hydroxy part. For this one could use chain elongation and other functional group diversification, or one synthesize more derivatives of **1g**. It is also important to note that the nitro group in **1a** is a mandatory part of the pharmacophore. This limits structural variations of the heterocyclic core of **1a**, because of the difficulty of synthesizing highly functionalized 5-nitrofuran-2-carboxamides.

Regarding the analogs of **2a**, moderate to no activity was mostly observed compared to the parent hit. 3,4-di-O-caffeoyl-γ quinide lactones (as the non-acetylated derivatives of **2f** and **2g**) are known natural compounds, isolated from some edible plants and members of the family of chlorogenic acids. 4-deoxy quinates (**2a**–**e**, **2h**,**i**) are original synthetic compounds, prepared as more stable analogs of the natural compound 3,5-di-O-caffeoyl quinic acid by preventing the known migration of the cinnamoyl moieties. These preliminary results reveal that the presence of two caffeoyl residues is required whatever the nucleus, 4-deoxy quinic acid, or γ quinide lactone, albeit active only at 100 µM in the latter case (**2f**). The absence (**2i**), the removal of one caffeoyl group (**2g**), or the partial or complete replacement by a coumaroyl residue (**2b**–**e**) lead to complete loss of activity. The hydrolysis of methyl quinate into carboxylic acid (**2h**) also cancelled the positive response observed for **2a**. Although composed of homodiesters, but also heterodiesters in their quinic acid or in their methyl quinate form, and of two γ quinide lactones, a larger structural diversity could benefit from a more thorough SAR study.

### 3.4. Comparison of the NMD Inhibition Efficacy of Molecules **1a** and **2a** versus Other NMD Inhibitors

In order to evaluate the potency of molecules **1a** and **2a** as NMD inhibitors, these molecules were added with increasing concentrations to the culture medium of U2OS cells expressing the construct described in Figure 2A. The NMD inhibition curves obtained were then compared with the NMD inhibition levels obtained in the presence of previously reported NMD inhibitors such as NMDI-14, colchicine, and amlexanox (Figure 3) [21,22,29]. We present the results of two representative experiments rather than plotting an average, because basic luciferase activity was highly variable from one experiment to another (variations in the number of cells and transfection rate precluded proper calculation of an average curve). All the molecules tested showed an ability to inhibit NMD, although the potency of NMDI-14 appeared very limited (to be fair, this certainly stemmed from the fact that we failed to resuspend this molecule properly). Amlexanox and colchicine showed the maximum inhibition of NMD from the first concentration tested; then, a plateau was reached. In the case of amlexanox, the plateau was followed by a decrease in luciferase activity, whereas colchicine retained the same ability to inhibit NMD at all higher tested concentrations. These results suggest that both of these molecules can be used with this substrate to promote NMD inhibition at concentrations below 0.4 µM. The inhibition of the NMD observed amounted the background noise to about 2 to 3 times. Molecules **1a** and **2a** behaved somewhat differently: we observed a bell-shaped curve with a gradual rise, a peak, and a decrease. This drop might either reflect toxicity or decreased solubility, as we have already seen with other molecules such as 2,6-diaminopurine [28]. At their respective peaks, molecules **2a** and 1a molecules displayed an NMD-inhibiting potency quite similar to those of amlexanox and colchicine, although molecule 1a seemed slightly more potent than the other compounds. As maximum luciferase activity was observed around 6.2 µM for **1a** and 12.5 µM for **2a**, these are the working concentrations used throughout the remainder of this study.

### 3.5. Evaluation of NMD Inhibitor Molecule Cytotoxicity

The molecules we seek must inhibit NMD without inducing cell death if we hope to be able to exploit them in a therapeutic approach for genetic diseases. The cytotoxicity of the different NMD inhibitors shown in Figure 3 was evaluated at concentrations near their working concentrations. For this, adenylate kinase activity was measured in the culture medium of 16HBE14o- cells incubated with each NMD inhibitor separately (Figure 4). These cells were used because they are not cancer cells but cells which have been transformed by introducing a plasmid expressing the big T antigen [30]. They are therefore more sensitive to a potential toxic effect than tumor cells, which very often contain several mutations, particularly in tumor suppressor genes linked to DNA repair or cell cycle control mechanisms. Cycloheximide was used as a positive control because it is well known to cause the strong inhibition of NMD by blocking translation, and this induces cell toxicity. Among the five NMD inhibitor molecules tested, only colchicine showed significant toxicity. This was expected, because colchicine is a cytoskeleton disruptor. The toxicity observed with NMDI-14 was probably due to the poor solubility of the molecule and not to proven toxicity. Interestingly, molecules **2a** and **1a** showed no significant toxicity, although the latter showed a trend towards increased toxicity, suggesting that at higher concentrations, its toxicity could become significant.

### 3.6. Inhibition of Endogenous NMD by Molecules **1a** and **2a**

To validate the NMD-inhibiting capacity of molecules **1a** and **2a**, Calu-6 lung cancer cells carrying an NMD-activating nonsense mutation in the *TP53* gene (leading to NMD of the corresponding mRNA) were incubated with each molecule at concentrations near its working concentration. Cycloheximide was used as a positive control. The results are shown in Figure 5. Molecules **1a** and **2a** proved able to induce an increase in the level of p53 mRNA, reflecting NMD inhibition. The level of NMD inhibition attained was very similar to that reached by cycloheximide, a potent NMD inhibitor. Both of these new molecules thus appear as powerful inhibitors of NMD.

## 4. Conclusions

Here, we present a novel screening system for identifying NMD inhibitors based on a test in which a luciferase-encoding pre-messenger RNA carries an intron in its 3′UTR in order to target the corresponding mRNA for NMD degradation. Any NMD-inhibiting molecule will thus cause an increase in luciferase activity as compared to a situation in which NMD is not inhibited. We have validated this screening system by showing the ability of two families of molecules to inhibit NMD as effectively as cycloheximide, an NMD inhibitor recognized as very effective.

## 5. Discussion

Using an already proven two-plasmid screening system in which a firefly luciferase mRNA is subject to NMD because of UPF1 factor targeting its 3′UTR, we have selected two new molecules, called **1a** and **2a**, as potential NMD inhibitors (Figure 1). We have confirmed the inhibitory action of both molecules with a new, one-plasmid screening system in which a construct is subject to NMD because of the presence of an intron in the 3′UTR, transforming the physiological stop codon into a PTC (Figure 2A). This one-plasmid screening system has the advantage of being more physiological, since this situation is encountered naturally in certain mRNAs [31,32]. In addition, transfection bias is limited by the fact that only one plasmid is required. With this one-plasmid screening system, we have both validated the NMD-inhibiting action of molecules **1a** and **2a** and shown that derivatives of these molecules also exert such action. Our SAR study has made it possible to propose a minimal skeleton likely to be predisposed to being an NMD inhibitor for 1a-family molecules (Figure 2). For the **2a** family, it will be necessary to synthesize more analogues in order to refine the definition of a minimal structural skeleton. We have further compared the NMD-inhibiting efficacy of molecules **1a** and **2a** with those of other inhibitors previously identified and validated by us or other teams [19,21,22,29]. It emerges that both new NMD inhibitors can impact NMD at least as effectively as the other inhibitors (Figure 3), without showing significant toxicity at the concentrations causing the strongest inhibition of NMD (Figure 4). Strong NMD inhibition and low toxicity are two essential criteria for the identification and development of molecules liable to be used therapeutically, and NMD inhibitors are indeed viewed as potential candidates in the development of treatments for genetic diseases caused by a nonsense mutation. Nonsense mutations generally activate the NMD mRNA surveillance system, thus inducing the degradation of the mutant mRNA. Inhibiting NMD can have two effects, both of which are favored by increasing the amount of nonsense-mutation-carrying mRNA. On the one hand, this mRNA can be translated into a truncated protein which, depending on the position of the nonsense mutation, might partially or totally retain the function of the wild-type protein. In this situation, the NMD inhibitor is sufficient on its own as a potential tool for developing a therapeutic approach. On the other hand, the mRNA carrying the nonsense mutation constitutes a substrate for forced readthrough, a therapeutic approach in which the ribosome is forced to recognize the PTC as a codon and thus synthesize a full-length protein with, at most, one different amino acid as compared to the wild-type protein [14,33]. We have illustrated the potential of molecules **1a** and **2a** to inhibit NMD in a pathophysiological context by using lung cancer cells carrying a nonsense mutation in the tumor suppressor gene *TP53* (Figure 5). The inhibition of NMD is potentially a particularly interesting approach in the development of anti-cancer therapies, since it could make it possible to both express tumor suppressor genes carrying nonsense mutations or ones that are not expressed due to a regulatory gene undergoing NMD and also induce the expression of neo-epitopes [15,32,34]. Molecules **1a** and **2a** clearly appear capable of inhibiting NMD and thus increasing the amount of p53 mRNA in Calu-6 cells. Future studies will aim to assess the ability of these molecules to induce the synthesis of a functional truncated protein in an adapted pathological model. Another goal will be to find a compatible readthrough molecule for improving a readthrough-based therapeutic approach. It will also be essential to determine the mechanism through which these molecules inhibit NMD in order to design more effective molecules and also, certainly, improve our knowledge of the PTC recognition mechanism during NMD.

Although the therapeutic benefit seems undeniable, vigilance remains necessary, as NMD is not only a quality control mechanism but also a regulatory pathway for approximately 5 to 10% of human genes [32]. Gene deregulation could lead to the expression of aberrant metabolic pathways or even to cellular toxicity. The benefit–risk ratio must therefore be evaluated when an inhibitor is considered a potential drug candidate. This does not mean that NMD inhibition should be dismissed as a therapeutic approach, since amlexanox, a drug used in the treatment of mouth ulcers and certain forms of asthma, has been shown to inhibit NMD [21]. The impact of NMD inhibition on the human transcriptome might depend on the strength of the NMD inhibition caused by the molecule considered. One should remember that studies analyzing the impact of NMD on gene expression were carried out with siRNAs directed against NMD factors, causing very strong NMD inhibition. Such a level of inhibition should not be achieved with the small NMD-inhibiting molecules identified so far. Yet for the moment, no clinical trial aimed at inhibiting NMD has been initiated, probably due to the lack of NMD inhibitors meeting the efficacy and safety criteria for its administration to humans.

## Figures and Tables

**Figure 1 biomedicines-11-02801-f001:**
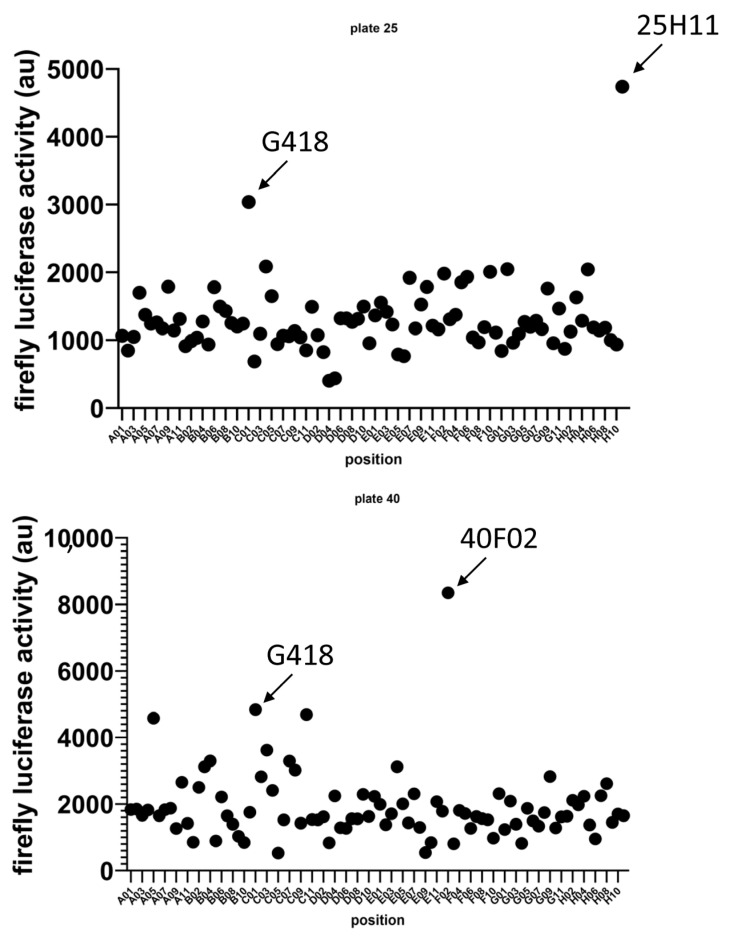
Identification via screening of two new potential NMD inhibitors. Scatter plot of luciferase activities measured in the different wells. Luciferase activity measurements related to incubation with molecules **1a** and **2a** are indicated by arrows. G418-related luciferase activity is also shown. G418 serves as positive control. The higher the luciferase activity, the greater the NMD-inhibiting potential of the studied compound.

**Figure 2 biomedicines-11-02801-f002:**
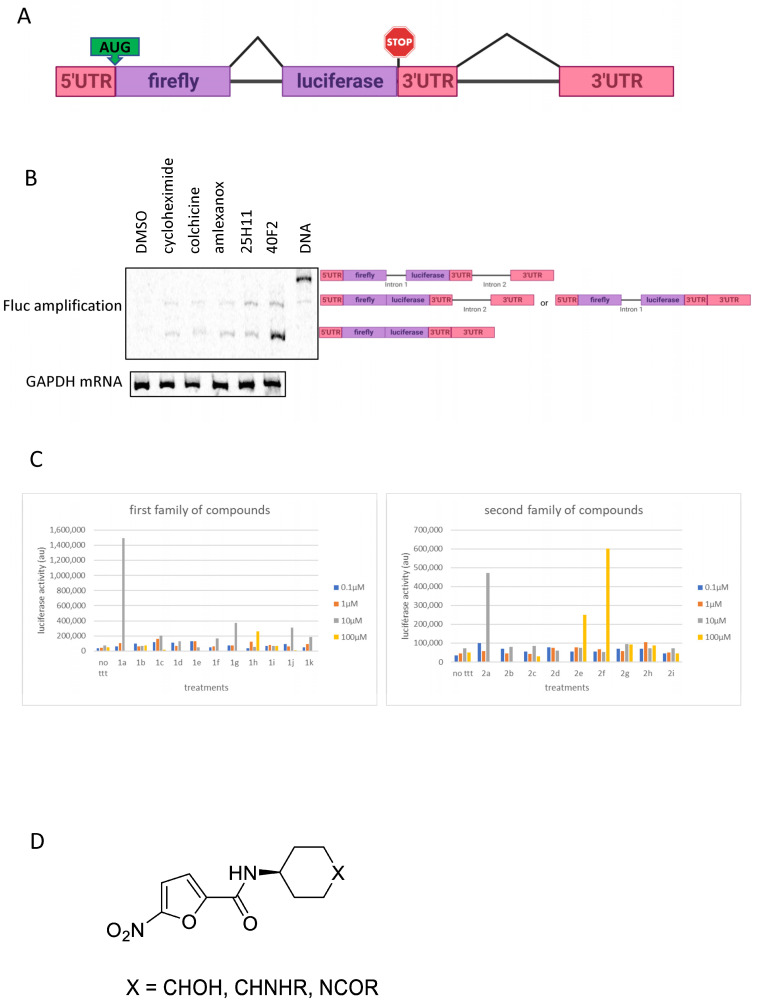
The one-plasmid screening system for identifying NMD inhibitors. (**A**) Schematic representation of the construct used to generate an mRNA subject to NMD because of the presence of an intron in the 3′UTR. Exons are represented by rectangles and introns by lines. The 5′UTR and 3′UTR are in red. The firefly luciferase reading frame is in purple. The AUG initiation codon and the STOP codon are indicated. (**B**) Verification of proper construct splicing. Analysis of firefly luciferase DNA amplification from the DNA by radioactive PCR (rightmost lane) and from the mRNA by radioactive RT-PCR. The different isoforms are indicated to the right of the gel. (**C**) Bar plot showing the extent of NMD inhibition (luciferase activity) measured in U2OS cells transfected with the construct described in A and incubated with derivatives of molecule **1a** (**left**) or molecule **2a** (**right**). (**D**) Minimal structural skeleton of 1a-family derivatives liable to promote NMD-inhibiting activity.

**Figure 3 biomedicines-11-02801-f003:**
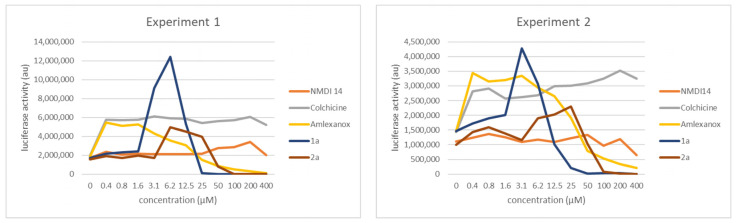
Compared NMD inhibition efficacies of different confirmed NMD inhibitors and molecules **1a** and **2a**. U2OS cells were transfected with the construct depicted in Figure 2A. Each molecule was tested at 0, 0.4, 0.8, 1.6, 3.1, 6.2, 12.5, 25, 50, 100, 200, and 400 µM to establish a dose–response curve. The results presented are representative of two independent experiments.

**Figure 4 biomedicines-11-02801-f004:**
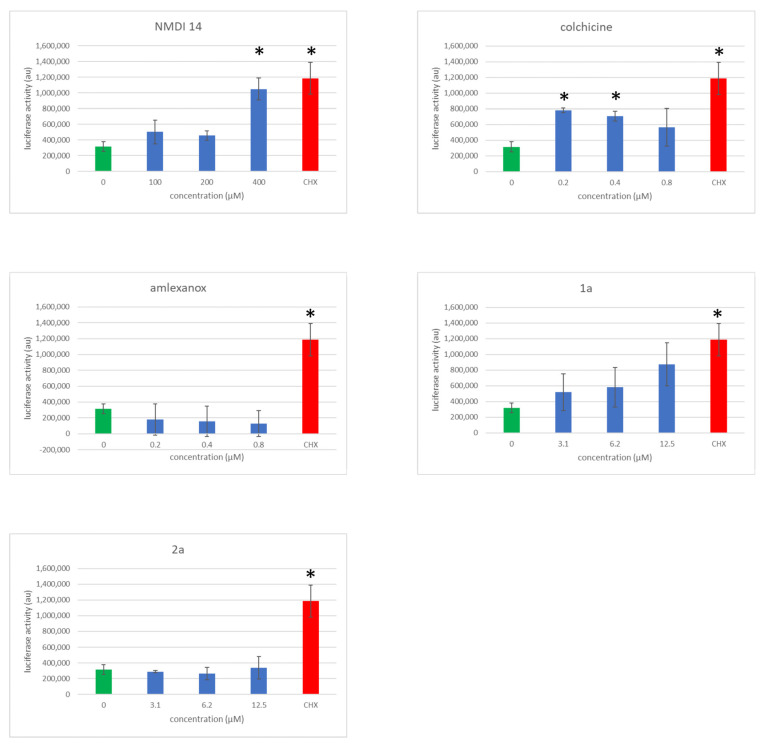
Molecules **1a** and **2a** show no significant toxicity at the doses tested. Adenylate kinase activity was measured in the culture supernatant of 16HBE14o- cells after 24 h of exposure to different concentrations of each molecule separately (working concentration, half the working concentration, and twice the working concentration). Cycloheximide was used at 0.5 mg/mL as a positive control. The results shown are representative of two independent experiments. *p*-values were calculated with Student’s *t*-test (* < 0.05). (red bar). Green bar corresponds to the no treatment condition (0).

**Figure 5 biomedicines-11-02801-f005:**
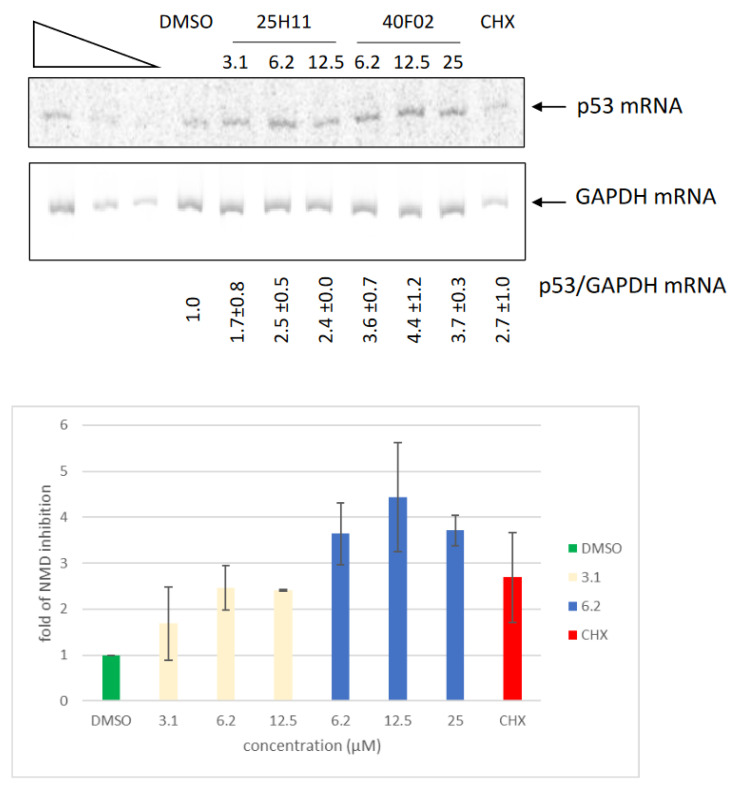
Molecules **1a** and **2a** inhibit NMD of endogenous p53 mRNA in Calu-6 cells. Calu-6 cells were incubated with molecule **1a**, molecule **2a**, DMSO, or cycloheximide (CHX) as positive control at 0.5 mg/mL. The upper panel shows a quantitative RT-PCR analysis gel, and the lower panel shows the ratio of p53 mRNA to GAPDH mRNA, normalized with respect to the results obtained with DMSO. The results are representative of two independent experiments.

## Data Availability

Data will be freely shared under request.

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
