# Peer review of "Identifying Potent Nonsense-Mediated mRNA Decay Inhibitors with a Novel Screening System"

_biomedicines, 2023, doi:10.3390/biomedicines11102801_

Round 1

Reviewer 1 Report

This paper is technically very good and very timely. There is a growing sense, particularly in rare diseases, that targeting nonsense-mediated decay inhibitors will be an important therapeutic target. This reviewer has no concerns on the science presented, however the authors need to take a broader perspective in regard to the significance of their results. A recent publication by Zanello et al in EMBO Molecular Medicine highlighted the need to target diseases with similar molecular etiology.  This current paper fits into this approach very much so.  Thus, in the introduction and the discussion, the authors could do a better job of highlighting the broad application of the inhibitors they have identified. One other minor note, some discussion is required on the potential of-target effects of the compounds as well as toxicity.

Reviewer 2 Report

Carrad et al described the use of a novel screening system to identify nonsense-mediated mRNA decay inhibitors. The manuscript is well organized, the results are interesting, the evidence is strong and the conclusions are useful. 

I only suggest to improve the abstract. Results and conclusion should be elucideted in the abstract. 

A paragraph "conclusions" could improve the clarity of the manuscript
